# Self-Assembled TiN-Metal Nanocomposites Integrated on Flexible Mica Substrates towards Flexible Devices

**DOI:** 10.3390/s24154863

**Published:** 2024-07-26

**Authors:** Juncheng Liu, Yizhi Zhang, Hongyi Dou, Benson Kunhung Tsai, Abhijeet Choudhury, Haiyan Wang

**Affiliations:** 1School of Materials Engineering, Purdue University, West Lafayette, IN 47907, USAtsai178@purdue.edu (B.K.T.); choudh54@purdue.edu (A.C.); 2School of Electrical and Computer Engineering, Purdue University, West Lafayette, IN 47907, USA

**Keywords:** plasmonic, mica, flexible devices, nitride-metal nanocomposite, sensors

## Abstract

The integration of nanocomposite thin films with combined multifunctionalities on flexible substrates is desired for flexible device design and applications. For example, combined plasmonic and magnetic properties could lead to unique optical switchable magnetic devices and sensors. In this work, a multiphase TiN-Au-Ni nanocomposite system with core–shell-like Au-Ni nanopillars embedded in a TiN matrix has been demonstrated on flexible mica substrates. The three-phase nanocomposite film has been compared with its single metal nanocomposite counterparts, i.e., TiN-Au and TiN-Ni. Magnetic measurement results suggest that both TiN-Au-Ni/mica and TiN-Ni/mica present room-temperature ferromagnetic property. Tunable plasmonic property has been achieved by varying the metallic component of the nanocomposite films. The cyclic bending test was performed to verify the property reliability of the flexible nanocomposite thin films upon bending. This work opens a new path for integrating complex nitride-based nanocomposite designs on mica towards multifunctional flexible nanodevice applications.

## 1. Introduction

Metamaterials with plasmonic nanostructures have gained great interest for various applications including biosensors, surface-enhanced Raman spectroscopy (SERS), and surface-enhanced infrared absorption spectroscopy (SEIRA) [1,2,3,4,5,6,7]. Noble metals, including Au and Ag, have been proven to be excellent candidates for such nanostructure designs due to their strong surface-enhanced plasmonic resonance from the visible to near-infrared range [8,9,10,11]. Various nanostructures have been developed, including nanopillars, nanoparticles, and nanoholes for high-resolution imaging [12,13,14,15,16]. Work on creating new nanostructure designs and optimizing the existing ones is crucial for SERS- and SEIRA-based sensors. Meanwhile, reducing optical loss and enhancing structural stability could further improve efficiency [15]. As alternative plasmonic candidates, transition metal nitrides, such as TiN, have been broadly studied for their plasmonic properties and potentials as nanostructured metamaterials with reduced optical losses [17,18,19]. Integrating optical metamaterials with other magnetic-based materials not only opens new platforms towards probabilistic computing and other combined device functionalities, but also improves the sensitivity and reproducibility for SERS [17,20,21,22].

In recent years, researchers have been focusing on finding pathways for the fabrication of wearable sensors for health monitoring. Currently, most medical diagnostic tools require drawing blood and traditional bench-top assay methods [23]. The use of flexible, non-invasive, and wearable biosensors for detecting biomolecules in biofluids can provide crucial insights into humans’ physiological condition, leading them to receive considerable attention from the personal healthcare industry [24]. Due to its non-invasive nature and sensitivity, SERS is a promising detection method to detect analyte in biofluids such as sweat [25,26,27]. This gives it an edge over the conventional electrochemical method [25,28]. Currently, for SERS application, most studies focus on various plasmonic nanomaterials. Magnetoplasmonic materials involve a combination of both magnetic and plasmonic properties of the materials [29]. As such, the use of such materials can provide an alternative solution to enhance the Raman signal under a magnetic field, thereby increasing the sensitivity of the sensor. The sensor architecture also needs to be designed for wearable systems, which necessitates the substrate to be robust enough to withstand multiple bending cycles.

Most plasmonic nanostructures and metamaterials have been fabricated using either top-down approaches, including e-beam lithography and the focused ion beam (FIB) method, or bottom-up methods, such as anodized alumina template and e-beam/FIB direct writing [30,31,32,33,34]. As a newly developed alternative, self-assembled nanocomposite thin films have been demonstrated for nanoscale metamaterial designs and processing [35,36]. A unique nanocomposite type in a pillar-in-matrix form, also called vertically aligned nanocomposites (VANs), provides opportunities in achieving strong anisotropy in physical properties, easy tuning of physical properties as well as combined functionalities. Considering phase compositions, oxide–oxide-based VAN systems have been widely studied [37,38,39,40,41], and oxide–metal VAN systems have been recently developed [35,42,43,44,45,46,47,48]. Nitride-based VAN systems are even more scarce [47,49,50,51,52]. Interestingly, most of the VAN demonstrations have been two-phase systems, with limited demonstrations in three phase systems [53,54,55] and alloyed metal pillars [56,57,58]. This is largely due to the narrow processing windows allowed for the three-phase co-deposition and epitaxial growth of three phases.

Among the previous reports of nitride-based VAN systems, one of the first demonstrations showed a novel two-phase Au-TiN-based nanostructure with tunable Au pillar density, leading to adjustable optical properties [49]. Another successful demonstration showed tailorable Ag nanopillar tilting in a TiN matrix to achieve anisotropic optical properties [50]. Co-deposition of a three-phase system consisting of a NiO core encased in a plasmonic Au shell within a TiN matrix capable of strong magneto-optical coupling has also been reported [53]. Other reports include ferromagnetic CoFe_2_ and Au–CoFe_2_ core–shell nanopillars embedded in a TiN matrix as multifunctional hybrid metamaterials [59,60]. However, other such TiN-based systems need to be explored and studied for the incorporation of these interesting systems in practical applications.

From the substrate selection point of view, most of the prior metamaterial demonstrations have focused on rigid substrates, such as Si, sapphire, MgO, and other single crystalline oxide substrates. Integrating plasmonic metamaterials on flexible substrates could be very interesting considering the needs on multifunctional materials for flexible electronics and photonics [28,61,62,63,64]. In general, three major classes of flexible substrates have been used for flexible device fabrication. Polymers such as polyethylene terephthalate (PET), polydimethylsiloxane (PDMS), and polyimide (PI) are the most widely used type due to their excellent elasticity and flexibility as well as low cost. However, the poor thermal stability of polymers limits their usage in the epitaxy growth of thin films. Wafer transfer could be a viable solution, while multiple steps are usually required, which will increase the fabrication cost [65]. Metal foils, including Ni and Al, are alternative choices to polymers, since they are highly thermally stable with a crystallized structure, which is ideal for thin film epitaxy growth [66,67,68]. However, metal foils typically require a very thick diffusion barrier layer to prevent metal diffusion during high-temperature growth. Layered materials, such as mica, have been developed as flexible substrates for thin film integration [69,70]. Mica is thermally stable, inexpensive, and biocompatible, with good flexibility and mechanical properties, which makes mica capable of epitaxy growth for thin films using most deposition techniques. Multiple oxide materials have been integrated on mica for different functionalities [71,72,73,74,75,76]. Hu et. al. demonstrated that the plasmon resonance on their mica heterostructures remained constant despite several bending cycles. These make mica an excellent substrate candidate for photonic devices such as wearable sensors. Recently, nanocomposite integration on mica has also been realized, including several oxide–oxide and oxide–metal nanocomposites [77,78,79,80]. Up to now, no prior study on transition metal nitride-based nanocomposite thin films on mica substrates has been demonstrated.

In this work, three nitride–metal nanocomposite thin films, i.e., TiN-Ni, TiN-Au, and TiN-Au-Ni, have been integrated on mica substrates using PLD, as illustrated in Figure 1. TiN is selected considering its high thermal/chemical stability, plasmonic properties, and low-loss nature to serve as a matrix material for the nanocomposite, with Ni as a typical ferromagnetic metal and Au as a plasmonic noble metal. The combined plasmonic and ferromagnetic properties are desired in the newly designed TiN-Ni and TiN-Au-Ni nanocomposite systems, with the goal also being to explore the growth capability and mechanism of complex three-phase nanocomposites on mica. Structural characterization was conducted to understand the microstructure of all the composite films. Optical and magnetic properties were measured and compared. The bending test was performed to verify the stability of the integrated VAN films upon cyclic bending.

## 2. Results and Discussion

Pulsed laser deposition with a KrF laser (Lambda Physik Compex Pro 205, λ = 248 nm) (Coherent Corp.—Global, Saxonburg, PA, USA) was used to deposit the nanocomposite films on the mica (001) substrate. A commercial TiN target with Au and Ni strips was used for the deposition. The substrate temperature was 600 °C during deposition. To investigate the crystallinity information of the nanocomposites, an X-ray diffraction (XRD) θ-2θ scan was carried out for all three films. XRD was performed using a Panalytical X’Pert X-ray diffractometer (Malvern Panalytical, Malvern, UK). As shown in Figure 2, for TiN-Ni/mica film, TiN (111) and (222) peaks can be identified, while Ni also grows in the (111) direction. For the TiN-Au/mica film, both TiN (111) and (200) peaks are observable, with (111) as the dominate growth orientation for TiN. Au (111) growth can also be found in the TiN-Au/mica film, with partial overlap with the TiN (111) peak. The (111) growth of the matrix and the metals in the TiN-Au/mica and TiN-Ni/mica films is coherent with the three-fold symmetry of mica (001). Interestingly, for the co-growth TiN-Au-Ni/mica film, only the TiN (200) peak can be seen as the dominant growth orientation for TiN. The Au (200) and Ni (111) peaks could potentially overlap with the mica (001) peak. Overall, all three films show the high crystallinity of the film phases.

To understand the overall microstructures and morphology of the nanocomposite films, scanning transmission electron microscopy (STEM) was conducted on the TiN-Au/mica and TiN-Ni/mica films. The TEM, STEM, and EDX were conducted on FEI TALOS F200X (Thermo Fisher Scientific, Waltham, MA, USA). The deposited film was taken for TEM sample preparation, achieved by conducting hand polishing and grinding manually, until a desired thickness. Later, the sample was then taken for dimpling and polishing in diamond paste. Finally, the sample was kept for ion milling to obtain a thin area for TEM. As shown in Figure 3a, a clear VAN structure can be found for the TiN-Au/mica film, and Au grow as pillars within the TiN matrix. The average diameter of the Au pillars is ~ 4 nm. For the TiN-Ni/mica film, as shown in Figure 3b, a pillar-in-matrix structure can also be seen, and Ni grow as pillars embedded in the TiN matrix. The diameter of the Ni pillars is around 2 nm. Interestingly, both Au and Ni pillars in both films are slightly tilted (~10° for the TiN-Au system and ~25° for the TiN-Ni system), which could be related to the texture of (111) and the tendency to expose the low-surface-energy facets of (111) by tilting the (111) oriented pillars.

The microstructure analysis was also performed on the TiN-Au-Ni/mica film, as summarized in Figure 4. It is interesting that the co-growth of the three phases results in a vertical pillar growth in the matrix, with the Au and Ni phases forming a core–shell-like pillar structure, i.e., an Au-core and Ni-shell, embedded in the TiN matrix, as suggested by both the cross-sectional and plan-view STEM images as well as the corresponding EDS mappings. More specifically, based on the composition EDS mapping in Figure 4m, Au prefers to grow in the inner part of the pillars, while Ni shows a higher distribution at the shell of the pillar. It is possible that there is a transition layer at the shell of the pillars with Au and Ni alloyed together. The co-growth of the two metals shows more vertical pillars, which are related to the preferred (002) texture of Au based on XRD (Figure 2) and TEM plan-view (Figure 4h), and this is also typical for other Au-alloyed metal pillar growths in VANs [50].

Considering the ferromagnetic nature of Ni, magnetic property measurement was conducted on the TiN-Ni/mica and TiN-Au-Ni/mica films. A MPMS-3 SQUID Magnetometer (Quantum Design, San Diego, U.S.) was used for this characterization. Both the in-plane (IP) and out-of-plane (OP) measurement results for both films are plotted in Figure 5a,b. It is clear that both films show a soft ferromagnetic response to the applied magnetic field. For the TiN-Ni/mica film, the coercive fields for the IP and OP are 77.0 Oe and 45.8 Oe, with corresponding saturation moments of 96.7 emu/cm^3^ and 84.8 emu/cm^3^. For the TiN-Au-Ni/mica film, the coercive fields and saturation moments for the IP and OP are 102.5 Oe, 5.8 emu/cm^3^, and 144.3 Oe, 5.4 emu/cm^3^. The low coercive field of the TiN-Ni/mica film is owing to the very small diameter of the Ni nanopillars. The TiN-Au-Ni/mica film shows a higher coercive field in the OP, with a similar saturation moment in both directions. The out-of-plane-dominated anisotropic magnetic property of the TiN-Au-Ni/mica film is mainly due to the structural anisotropy. The overall soft saturation moment of the TiN-Au-Ni/mica films is mainly due to the very thin shell structure of Ni in the system. Note that the volume used to calculate the moment was the entire film volume. The actual magnetization of Ni regions could be much higher.

Since the films were fabricated on flexible substrates, the stability of the film’s physical properties was verified against the bending test. Magnetic property measurement was also conducted on the TiN-Au-Ni/mica film when bent concavely, convexly, and after cyclic bending in both IP and OP directions. A stable magnetic response could be found in both directions, as shown in Figure 5c,d. The coercive fields stayed around 102.5 Oe and 144.3 Oe for IP and OP, and the saturation moments stayed around 5.8 emu/cm^3^ and 5.4 emu/cm^3^ at the different bending statuses and after 1000th bending. The robustness of the magnetic property suggests the high stability of the nitride-based nanocomposite films.

Optical measurements were also conducted to characterize the optical properties of the nitride nanocomposite films. Ellipsometry measurement was first performed to attain the permittivity of the films by fitting the ellipsometry parameters. A RC2 ellipsometer (J.A. Woollam, Lincoln, U.S.) was used for the ellipsometry measurement. The uniaxial B-spline mode was used for the fitting. As shown in Figure 6a,b, all films showed similar trends in both IP and OP directions, with the TiN-Ni/mica behaving as even more metallic. A transmittance measurement was also conducted, with the result plotted in Figure 6c. The transmittance measurement was performed on a UV–vis–NIR spectrophotometer (Perkin Elmer Lambda 1050). The typical plasmonic resonance ranges for TiN, Ni, and Au are 375 nm, 400–500 nm, and 500–600 nm, respectively, and the size of the nanostructure also impacts the plasmonic resonance wavelength. From the results, a red shift can be observed by varying the metallic component of the nanocomposite from Ni to Ni-Au and Au, which is due to the higher Au contribution to the overall plasmonic response of the film. Two major plasmonic peaks at around 470 nm and 515 nm could be observed in the TiN-Ni/mica and TiN-Au-Ni/mica film, with a higher absorption in the TiN-Au-Ni/mica film. The peaks become less obvious in the TiN-Au/mica film, probably due to a stronger plasmonic response near 485 nm from Au. The transmittance result of the TiN-Au/mica film is also consistent with a previous report of the same system demonstrated on an MgO substrate. The transmittance measurement was also performed for the TiN-Au-Ni/mica film when bent concavely and convexly, as shown in Figure 6d. However, the tilting of pillars might cause a minor shift in the transmittance peaks. The consistent film transmittance at different bending statuses indicates the reliability and robustness of the nanocomposite film. Note that the absolute value of transmittance difference between the films is mainly from their substrate thickness difference. From the Raman measurement in Appendix A, the intensity of the TiN-Au-Ni film is higher than the TiN-Au and TiN-Ni films. The Raman shift for the films at approximately 215 cm^−1^, 305 cm^−1^, and 580 cm^−1^ are related to the transverse acoustic, longitudinal acoustic, and transverse optical modes of TiN.

Overall, all three films possess a pillar-in-matrix microstructure. Both the TiN-Au-Ni/mica and TiN-Ni/mica film show a room-temperature ferromagnetic nature, with tunable plasmonic response observed in all three films. Compared to previous oxide-based VAN integrations on mica, the TiN-based VANs provide more opportunity for optical-related device applications due to their intrinsic plasmonic nature and high temperature stability. The mechanical flexibility of the films as well as the robust physical properties of the films are crucial for the design and application of future flexible SERS-based sensors. Using the above properties, a novel SERS-based sensor can be created. The use of mica allows for use of the sensors in more aggressive environments due to its high thermal and chemical stability. The use of a TiN matrix allows for use in more extreme environments and provides better protection to the core shell VANs as a matrix. Thus, a novel sensor structure has been proposed in Figure 7, consisting of TiN Au-Ni VANs as an active sensing layer on the flexible mica substrates, along with a power supply unit and a communication unit, all encased in an encapsulation layer.

For future directions, the magneto-optical coupling effect of the systems is worth exploring further, considering the plasmonic matrix and magnetic Ni pillars. Meanwhile, other transition metal nitride-based nanocomposites with different metal components, including Co, Au, Ag, and Fe, could also be explored on a mica substrate towards unique magneto-optical coupling demonstrations in nitride-based VAN systems. Adding the magnetic material to an optical metamaterial as a new platform for SERS could potentially increase the sensitivity.

## 3. Conclusions

Three different types of nitride–metal nanocomposite thin films, including TiN-Ni, TiN-Au, and TiN-Au-Ni systems, have been integrated on mica substrates. All the films present a pillar-in-matrix microstructure, with Au and Ni forming a unique core–shell-like pillar for the TiN-Au-Ni/mica film. A room-temperature ferromagnetic property was confirmed for both the TiN-Au-Ni/mica and TiN-Ni/mica films. A tunable transmittance property was realized by varying the metallic component of the nanocomposite film. The TiN-Au-Ni/mica film showed a higher absorption than the other two films by showing major plasmonic peaks at 470 nm and 515 nm. For the TiN-Au-Ni/mica film, the coercive fields and saturation moments for in-plane (IP) and out-of-plane (OP) configurations were 102.5 Oe, 5.8 emu/cm³, and 144.3 Oe, 5.4 emu/cm³, respectively. These values remained consistent, with coercive fields around 102.5 Oe for IP and 144.3 Oe for OP, and saturation moments around 5.8 emu/cm³ for IP and 5.4 emu/cm³ for OP, under various bending conditions and even after 1000 bending cycles. The physical properties were proven to be stable against the cyclic bending test. This work opens an avenue towards complex nitride-based nanocomposite designs for multifunctional flexible devices and sensor applications. Thus, this demonstration of a magneto-optical system on a robust mica substrate, which retains its properties despite undergoing several bending cycles, could provide a pathway for the development of future SERS-based wearable based sensors.

## Figures and Tables

**Figure 1 sensors-24-04863-f001:**
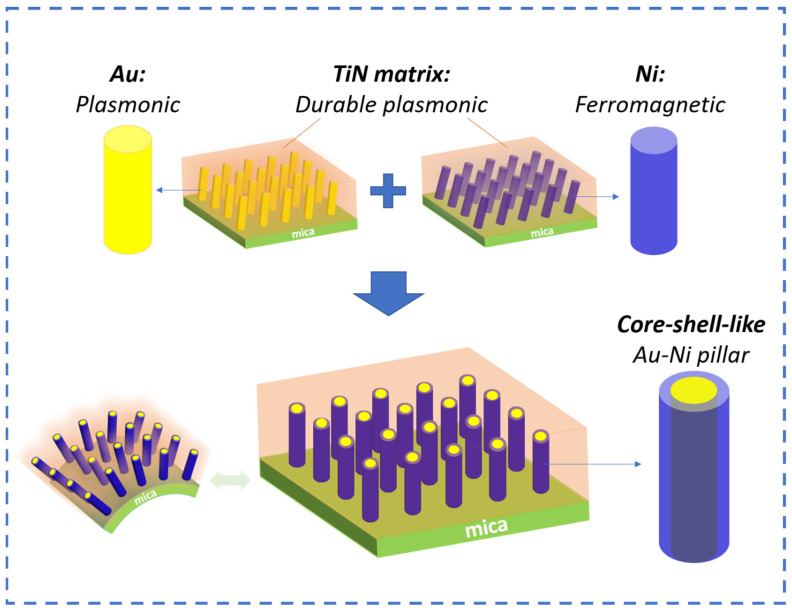
Conceptual drawing of the design of the work. TiN-based nanocomposite films with plasmonic Au pillars and ferromagnetic Ni pillars as well as core–shell-like pillars demonstrated on flexible mica substrates.

**Figure 2 sensors-24-04863-f002:**
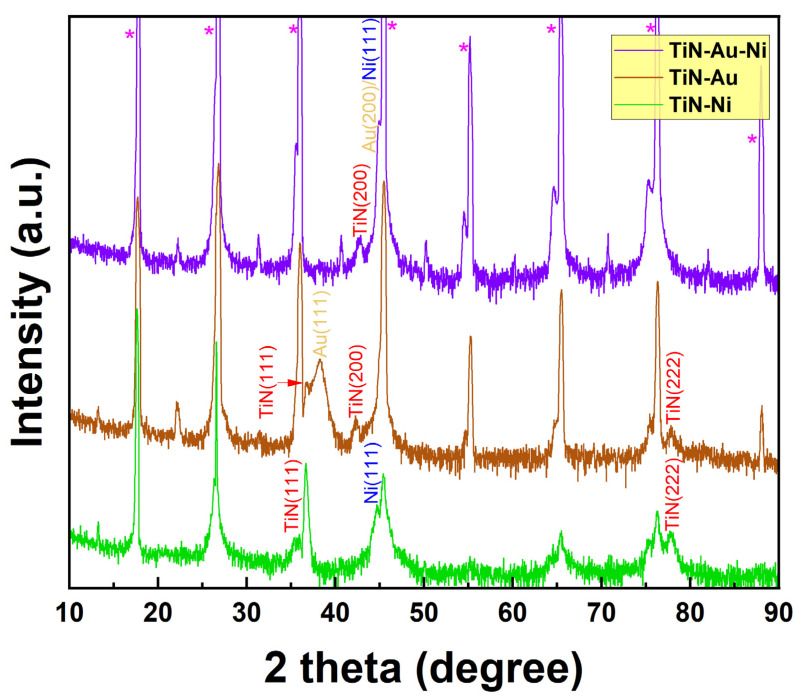
XRD results of the TiN-Ni/mica, TiN-Au/mica, and TiN-Au-Ni/mica films. * stands for mica (001) peaks.

**Figure 3 sensors-24-04863-f003:**
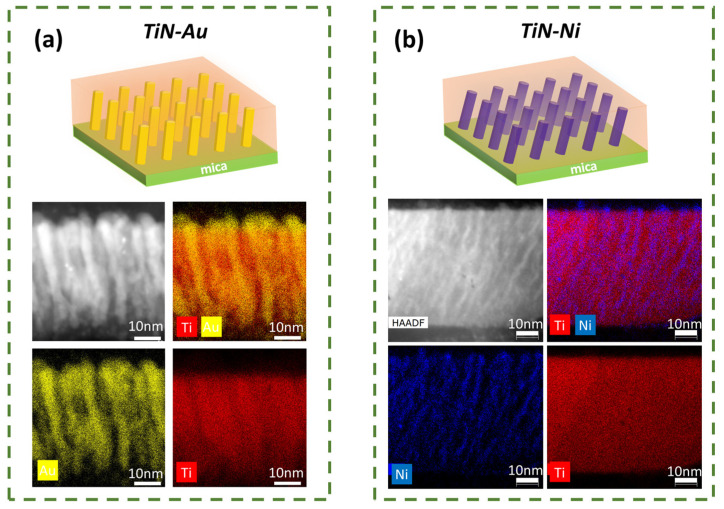
Schematic drawings and cross-sectional STEM image as well as the corresponding EDS mappings for (**a**) TiN-Au/mica and (**b**) TiN-Ni/mica films.

**Figure 4 sensors-24-04863-f004:**
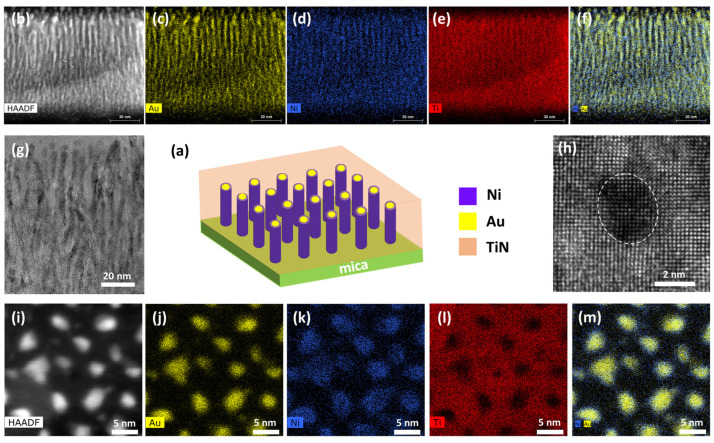
(**a**) Schematic drawing of TiN-Au-Ni/mica film. (**b**) Cross-sectional STEM image with (**c**–**f**) corresponding EDS mappings. (**g**) Cross-sectional TEM image. (**h**) Plan-view TEM image. (**i**) Plan-view STEM image, with (**j**–**m**) corresponding EDS mappings.

**Figure 5 sensors-24-04863-f005:**
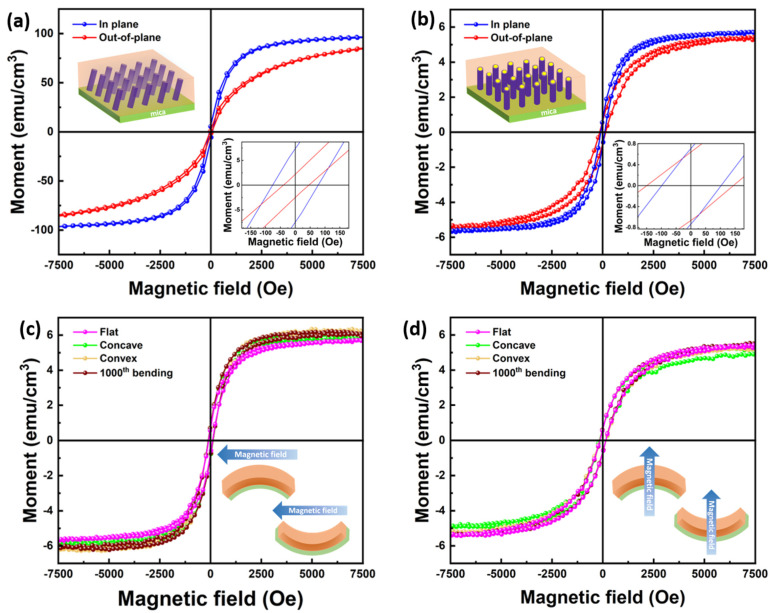
IP and OP *M-H* curves for the (**a**) TiN-Au-Ni/mica and (**b**) TiN-Ni/mica film. (**c**) IP and (**d**) OP *M-H* curves for the TiN-Au-Ni/mica film after the bending test.

**Figure 6 sensors-24-04863-f006:**
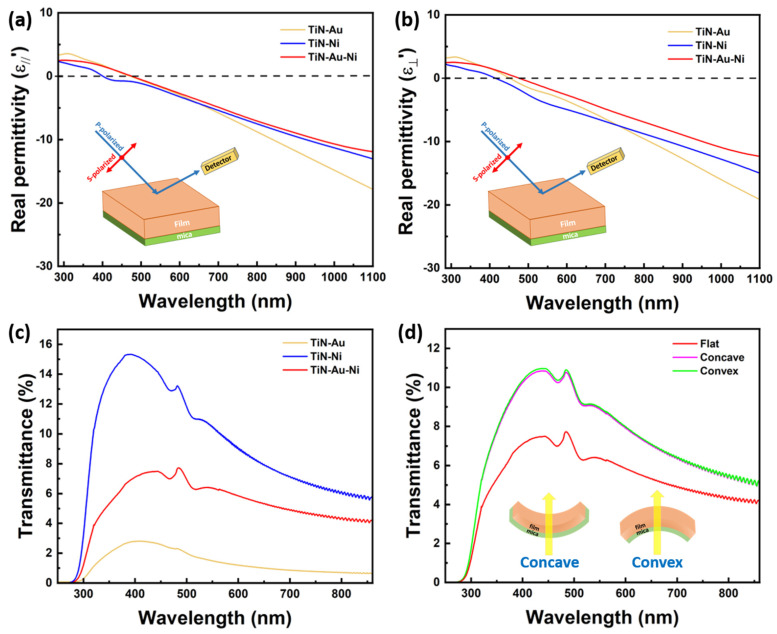
(**a**) IP and (**b**) OP real permittivity of the TiN-Au/mica, TiN-Ni/mica, and TiN-Au-Ni/mica films. (**c**) Transmittance of the three films. (**d**) Transmittance of the TiN-Au-Ni/mica film at different bending statuses.

**Figure 7 sensors-24-04863-f007:**
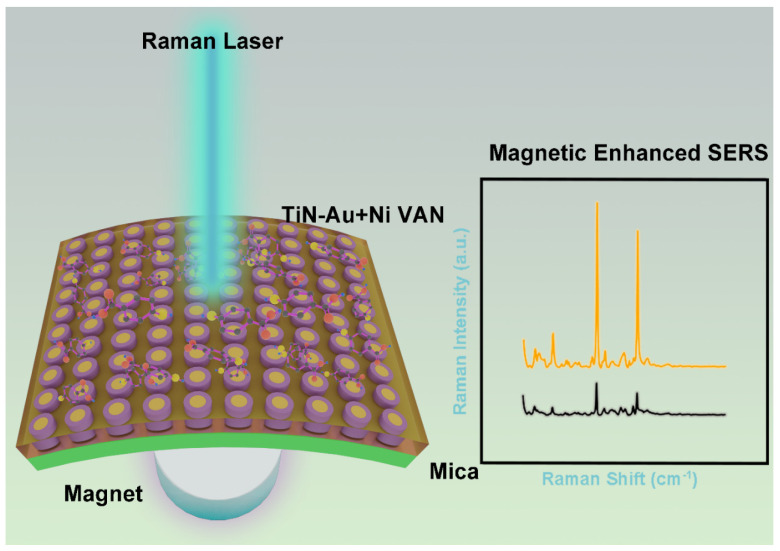
Proposed device on a magnetic field enhancing the Raman signal using the TiN-Au-Ni VANs on mica.

## Data Availability

Data is contained within the article.

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
