# Peer review of "Self-Assembled TiN-Metal Nanocomposites Integrated on Flexible Mica Substrates towards Flexible Devices"

_sensors, 2024, doi:10.3390/s24154863_

Round 1

Reviewer 1 Report

Comments and Suggestions for Authors

In the present work, a new path for integrating complex nitride-based nanocomposite was given. The microstructure evolution and properties of the composite film were characterized. It is a topic of interest to the researchers in the related areas but the manuscript needs improvement before acceptance for publication.

The comments are as follows:

(1) The main application of this composite film is as a SERS substrate, but the manuscript did not test the SERS performance. It is recommended to supplement.

(2) The details of sample preparation are unclear.

(3) There are some writing errors in punctuation and other aspects, which need to be carefully checked and corrected.

Author Response

Dear Reviewer

Reviewer 2 Report

Comments and Suggestions for Authors

In this manuscript, 3 different types of nitride-metal nanocomposite thin films, including TiN-Ni, TiN-Au and TiN-Au-Ni, on mica substrates have been fabricated and compared, which showed its potential applications on flexible or wearable sensors. I think this manuscript can be accepted for publication after minor revision.

1.      The authors have conducted a large amount of literature research, yet lacks a clear discussion on the innovative aspects and performance advantages of the TiN-metal nanocomposites devices compared to prior works.

2.      The Experimental Section would benefit from more detailed descriptions of device fabrication methods.

3.      There is no mention of whether the angular tilt of pillar structures has any impact on experimental results, which should be addressed.

Author Response

Dear Reviewer

Reviewer 3 Report

Comments and Suggestions for Authors

The author has presented an interesting work on the fabrication of flexible metasurface devices. It can be out of the scope of the journal if it does not demonstrate any sensing application. Therefore, I have the following suggestions to enhance the quality of the work. 

1) The sections related to the fabrication, microstructure investigation, magnetic property measurement, and optical measurement should be in the results and discussion section. The paper should finish with a conclusion. 

2) From STEM images, why it is not possible to see a clear structure of pillars deposited on a substrate?

3) The paper is interesting, however, it lacks some real-time applications such as sensing or imaging. Can the author perform some additional experiments to demonstrate its usability in any useful application? Otherwise, the paper is solely based on fabrication and its characterization results and it does not provide any sensing application which makes it out of the scope of the journal. 

4) The conclusion section is too general. I suggest adding the results in terms of obtained values such as transmittance of XX % or dB., etc. 

5) A comparison table should be added to the paper to compare the obtained results from the previous devices published in the literature.

Comments on the Quality of English Language

None. 

Round 2

Reviewer 3 Report

Comments and Suggestions for Authors

The author has not answered my query 3. I suggest the authors to read the question carefully and provide a satisfactory answer.

Comments on the Quality of English Language

None
